# Two-Dimensional Interfacial Exchange Diffusion Has the Potential to Augment Spatiotemporal Precision of Ca^2+^ Signaling

**DOI:** 10.3390/ijms23020850

**Published:** 2022-01-13

**Authors:** Cornelis van Breemen, Nicola Fameli, Klaus Groschner

**Affiliations:** 1Department of Anesthesiology, Pharmacology and Therapeutics, University of British Columbia, Vancouver, BC V6T 1Z3, Canada; 2Independent Researcher, Vancouver, BC V5Z 1R1, Canada; nicola.fameli@ubc.ca; 3Gottfried Schatz Research Center—Division of Biophysics, Medical University of Graz, 8036 Graz, Austria

**Keywords:** Ca^2+^ signaling, nano-junctions, intermembrane nano-spaces, Ca^2+^ transport, Ca^2+^ lipid interaction, surface diffusion

## Abstract

Nano-junctions between the endoplasmic reticulum and cytoplasmic surfaces of the plasma membrane and other organelles shape the spatiotemporal features of biological Ca^2+^ signals. Herein, we propose that 2D Ca^2+^ exchange diffusion on the negatively charged phospholipid surface lining nano-junctions participates in guiding Ca^2+^ from its source (channel or carrier) to its target (transport protein or enzyme). Evidence provided by in vitro Ca^2+^ flux experiments using an artificial phospholipid membrane is presented in support of the above proposed concept, and results from stochastic simulations of Ca^2+^ trajectories within nano-junctions are discussed in order to substantiate its possible requirements. Finally, we analyze recent literature on Ca^2+^ lipid interactions, which suggests that 2D interfacial Ca^2+^ diffusion may represent an important mechanism of signal transduction in biological systems characterized by high phospholipid surface to aqueous volume ratios.

## 1. Introduction

The discovery of ATP-driven accumulation of ionic calcium (Ca^2+^) by isolated vesicles of the endoplasmic reticulum (ER) of skeletal muscle in 1962 [1] established its fundamental role in cellular Ca^2+^ signaling. The presence of this physiologically important intracellular Ca^2+^ store was confirmed by the observation that an agonist induced a single, transient, smooth muscle contraction in the complete absence of external Ca^2+^, achieved by its removal from the bathing solution and displacement from extracellular binding sites by the tri-valent lanthanum cation [2]. However, it has become increasingly clear that Ca^2+^ signaling is much more complex than a simple stimulus response, as nearly all cellular and bodily functions are simultaneously and selectively regulated by ER-mediated Ca^2+^ transport. Currently, known Ca^2+^-sensitive functions include the following: contraction; relaxation; hyperpolarization; depolarization; ER refilling; ER unloading; secretion; endocytosis; protein folding; apoptosis; mitochondrial energetics; neurotransmitter release; intracellular trafficking; and intercellular communication via gap junctions. For this single ionic messenger to harmoniously control such a great range of biological mechanisms, it is crucial that its signals are delivered with pinpoint precision and millisecond timing. The ER is the main organelle that orchestrates this essential spatiotemporal precision of Ca^2+^ signaling. In order to accomplish this task, the cellular ER network is endowed with numerous different close-contact sites, termed nano-junctions (NJs), with the plasma membrane (PM), mitochondria, lysosomes and other organelles [3] as illustrated in Figure 1. Structural and functional aspects of NJs have been extensively studied in the past decade [4,5], while the molecular basis of information transfer by Ca^2+^ within these sites of unique membrane architecture still remain incompletely understood. 

NJs have been defined as cytoplasmic sub-compartments where membranes of different organelles appose each other within the nanometer scale. Typically, limiting membranes are separated by 20 nm or less and the specialized signaling function has been shown to fail at separation distances greater than 50 nm [6,7,8]. The main function of the NJs in cells is to precisely localize Ca^2+^ signals to specific Ca^2+^ sensors positioned within organellar-membrane or PM domains, while bypassing the bulk cytoplasm. The physiological mechanisms involved in a variety of different types of NJs, with one specific type for each different function, are as follows: (1) close apposition of Ca^2+^ source and target; (2) restricted Ca^2+^ diffusion within the nano-space; and (3) physical separation of different NJ types by a combination of spacing and buffer barriers to Ca^2+^ diffusion. For example, the process of refilling the endoplasmic/sarcoplasmic reticulum (SR) of vascular smooth muscle cells during asynchronous [Ca^2+^]_cyt_ oscillations is achieved by coupling Ca^2+^ entry via Ca^2+^-influx-mode NCX (rNCX) to SERCA at PM–SR NJs. The ultrastructure of the NJ optimizes SERCA Ca^2+^ uptake by reducing leakage into the bulk cytoplasm [9]. By employing stochastic particle simulator modeling software and using known NCX and SERCA turnover rates and surface densities, it was possible to generate a computational model of this cellular-signaling process, which demonstrated that the rate of Ca^2+^ entry via rNCX/SERCA was indeed sufficient for replacing the Ca^2+^ released by periodic opening and closing of inositol 1,4,5-trisphosphate receptors (IP_3_Rs) during the activation of asynchronous cytoplasmic Ca^2+^ waves that stimulate contraction [7]. However, in order to generate plausible predictions using our computational model, it was necessary to implement a Ca^2+^ target size on SERCA of approximately 20 nm^2^. This is 2500 times larger than the area occupied by the dehydrated Ca^2+^ with a diameter of 1Å. Assuming that short-range local electrostatic forces of attraction between fixed negatively charged binding sites on the SERCA macromolecule and the positively charged Ca^2+^ would increase the effective target size to an area several times larger than the size of the non-hydrated Ca^2+^, it would still be orders of magnitude smaller than 20 nm^2^. Therefore, in order to achieve effective functional transfer of Ca^2+^ from NCX on the PM to SERCA on the SR, it appears that an additional, yet ignored, mechanism is operative for supporting the linkage between Ca^2+^ signaling elements (sources and sinks) within NJs. To this end, all available computational-modeling data prompt us to conclude that, by itself, a 3D random walk of Ca^2+^ between its sources and sinks on the two closely apposing membrane surfaces may be insufficient for efficient NJ Ca^2+^ signaling. Thus, we propose that, in addition to the three NJ-related mechanisms mentioned above, a fourth requirement consists of 2D exchange diffusion of Ca^2+^ on the targeted phospholipid membrane surface.

## 2. Background

It is well established that Ca^2+^ can move rapidly through negatively charged solid lattices, such as fluorapatite, by the process of exchange diffusion [10]. An analogous model, featuring negatively charged phospholipids (PLs), has been described previously [11,12,13]. This experimental model consists of a Millipore filter impregnated with a mixture of phospholipids of animal origin, separating two aqueous phases, and it exhibits properties that are highly relevant to the topic of Ca^2+^ movements through narrow aqueous passages lined by PLs. Its salient feature is that it supports net transfer of Ca^2+^ through relatively long PL-lined pores at a much faster rate than would be possible for free diffusion within the limited adjoining aqueous phase. The mechanism that was presented to explain Ca^2+^ transport through this solid ion-exchange membrane involves the association of Ca^2+^ with a negatively charged phosphate or carboxyl group of the PL surface on the *cis*-side of the membrane, followed by transfer of Ca^2+^ within a 2D matrix of similar sites, constituted by pore-lining PL layers, and a final step of dissociation of Ca^2+^ from the negatively charged PL head groups on the *trans*-side of the membrane. The rate of this PL-mediated transport of radioactive-labeled Ca^2+^ through the membrane decreased by removal of Ca^2+^ from the buffered solution on the *trans*-side of the membrane. Paradoxically, further removal of the remaining Ca^2+^ from the *trans*-solution by the addition of the soluble but non-permeant chelator EDTA increased the rate of PL-mediated net Ca^2+^ transport across the PL-lined Millipore filter by more than one order of magnitude. The mechanism proposed to explain the latter observation is that dissociation of Ca^2+^ from PL head groups on the *trans*-side of the membrane is rate-limiting, and the addition of a freely diffusible, impermeant Ca^2+^ binding site on EDTA to the *trans*-side of the flux chamber facilitates the dissociation of PL-bound Ca^2+^. Once Ca^2+^ transported from the *cis*-side of the flux chamber has been chelated by EDTA, which is dissolved in a large volume of buffered solution on the *trans*-side, it cannot rebind to the PL membrane, but is replaced by the next Ca^2+^arriving from the *cis*-side. In this model system, electroneutrality is preserved by the movement of Mg^2+^ and monovalent cations in the opposite direction. The mechanism envisioned for the transfer of Ca^2+^ from the PL membrane to EDTA involves an intermediary step of partial dissociation from the negative PL site and simultaneous association with a carboxyl group of EDTA. We propose, herein, that a similar mechanism is involved in the transfer of Ca^2+^ bound to PL head groups of the membranes lining the NJ to its biological target site on the Ca^2+^-receptor protein.

## 3. Model Description

When we compare the artificial PL-mediated Ca^2+^ transport in the model described above with biological PL-lined nano-spaces, some striking parallels become obvious. For example, rod outer segments of the bovine eye feature a 15 nanometer-wide cytoplasmic phase between the intercalated discs, which stretch for several microns [14]. The lipid bilayers lining this narrow space contain 45% phosphatidylethanolamine, 36% phosphatidylcholine and 16% phosphatidylserine, calculated as a percentage of the total PL. At physiological pH, phosphatidylcholine and phosphatidylethanolamine are zwitterions, and phosphatidylserine has one net negative charge [14]. In this example, it was calculated that, of the Ca^2+^ released during stimulation, 90% to 99% was bound to the PL head groups of membranes lining the nano-spaces between the intercalated discs, which is similar to the high ratio of bound/free Ca^2+^ in the above model membrane. Although historically it has been accepted that such binding slows diffusion due to a drastic decrease in freely diffusible Ca^2+^ in the aqueous phase, it is also possible that PL-bound Ca^2+^ within the water–PL membrane interface of the nano-space continues its trajectory by the mechanism of 2D Ca^2+^ exchange diffusion. We, therefore, propose that the mechanism of 2D Ca^2+^/Mg^2+^, K^+^ exchange diffusion at the aqueous–phospholipid interfaces of NJs facilitates targeting Ca^2+^ receptors located on organellar membranes and the inner PM. 

Returning to the example of NCX-mediated SR refilling in vascular smooth muscle, we propose that Ca^2+^ enters the NJ via the reverse mode of NCX located in the junctional domain of the PM. It will then perform a 3D random walk and when it hits the negatively charged PL-membrane boundary of the junctional nano-space then proceeds along the lipid¬–water interface by a series of steps of reversible binding to negatively charged oxygen molecules for some variable time before being released back into the aqueous phase to resume its 3D random walk. After a number of such cycles, Ca^2+^ is envisioned to hit the ER-membrane surface in the proximity of its target protein, in this case SERCA, which can then be reached more effectively by 2D surface exchange diffusion than would be expected if it depended solely on 3D diffusion in the aqueous phase.

A simplified model for this mechanism, which incorporates the PL Ca^2+^ surface exchange diffusion, described for the artificial membrane [11], with the dynamic modeling of SR Ca^2+^ refilling [7], is illustrated in Figure 2. This model predicts that 2D surface exchange diffusion can facilitate Ca^2+^ in reaching its target by two means: (i) by augmenting the rate of Ca^2+^ movement towards its target and thus enhancing the speed and intensity of the signal and (ii) by decreasing the loss of Ca^2+^ at NJ edges, further increasing the speed as well as the selectivity of Ca^2+^ signaling.

## 4. Discussion

### 4.1. Rationale of the Proposed Model

Due to the novelty of this concept, there are currently no experimental or computational tests reported with respect to the biological implications of 2D Ca^2+^ exchange diffusion, specifically for its role in junctional Ca^2+^ signaling. On the other hand, non-Brownian interfacial diffusion has been intensively studied in physical chemistry, using state-of-the-art techniques of single-molecule tracking and dynamic modeling [15]. A complex process of “Continuous Time Random Walk” (CTRW), which contains elements of “flying, hopping and crawling” and combines random-walk 3D diffusion, adsorption and desorption as well as 2D surface diffusion, has been well documented. Elementary processes of intermittent, surface-delimited (2D) diffusion are considered important in molecular recognition and chemical sensing at interfacial surfaces. It is fascinating that CTRW-type motion resembles trajectories of biological organisms executing foraging behavior and could theoretically also be successfully applied to targeting physiological receptors by signaling molecules [13]. With respect to biological Ca^2+^ signaling, it is particularly relevant that the ultrastructure of the endoplasmic reticulum as well as its appositions to other organelles exhibit features such as tortuosity, binding affinity and high surface to aqueous volume ratios, a combination of which is a prerequisite for elementary processes of intermittent surface diffusion.

The physical chemistry of Ca^2+^ interactions with lipid bilayers has also been extensively studied and shows that Ca^2+^ binding can lower negative surface charge [13,16], cause clustering of PLs [16,17] and promote membrane fusion events as well as anchoring proteins in the lipid bilayer [18]. The structural effects of Ca^2+^ binding to PLs require millimolar concentrations of Ca^2+^, such as those observed in extracellular space and the ER lumen [1]. On the other hand, the cytoplasmic surfaces of cellular membranes are exposed to sub-micromolar Ca^2+^ concentrations, which are three orders of magnitude lower than the cytoplasmic concentration of Mg^2+^ and, at least under resting conditions, will contain minimal bound Ca^2+^. Nonetheless, phosphate, carboxyl and carbonyl groups are effective Ca^2+^ binders at low concentrations and will buffer Ca^2+^ during transient elevations typically associated with Ca^2+^-signaling events. Recent studies in model membranes suggest that Ca^2+^-induced PIP_2_ clustering and Ca^2+^ coordination within a hydrogen-bond network formed by PL head groups occur at membrane surfaces at micromolar divalent cation concentrations [19]. Hence, the generation of a dynamic cation-binding matrix at PL-cytoplasmic interfaces appears plausible for membrane regions endowed with a sufficient level of PLs and exposed to transient Ca^2+^ rises. 

In essence, the negatively charged cytoplasmic surface of NJs provides all the features required for 2D surface Ca^2+^ exchange diffusion, which could promote efficient coupling between Ca^2+^ sources and Ca^2+^ targets during activation. A parallel mechanism of lateral diffusion of Ca^2+^ tightly bound to PLs in a monolayer cannot be excluded at this time, although the lateral mobility of free PLs in cell membranes is limited to a diffusivity of approximately 10 µm^2^/s, which is further slowed down in complexes with divalent cations [17] and presumably even less mobile in the vicinity of signaling proteins [20]. On the other hand, a matrix of PL head groups bridged by divalent cations, as suggested for PL-rich domains around Ca^2+^ signaling complexes in the context of PL signaling [21], may provide the ideal conditions for efficient linkage of Ca^2+^ sources to downstream targets via 2D surface Ca^2+^ exchange diffusion. Support in favor of such a membrane-delimited transfer of Ca^2+^ signals is derived from current concepts of tight interactions between membrane phospholipids and Ca^2+^ transport proteins and the well-documented functional relevance of these interactions. In this context, it is intriguing that PLs are a prominent component of the “annular lipid” shells, which typically surround transmembrane proteins involved in Ca^2+^ handling [22,23,24]. The role of these annular lipids for cation transport is still incompletely understood, but has been proposed to include the accumulation of cations in the vicinity of the transport molecule to promote transport efficiency and specificity [25]. Recent detailed insights into the structure of Ca^2+^-signaling complexes by crystallography and single-particle cryo-EM studies identified “non-annular” lipids, including PLs, which protrude into fenestrations and crevices of integral membrane proteins to govern Ca^2+^ transport and Ca^2+^ regulatory or sensory functions [26,27,28,29]. Of note, such non-annular, “structural” PLs were found to reside within clefts of the SERCA complex to determine the handling of Ca^2+^ within the transporter [22]. Interestingly, for SERCA, a distinct “path structure” was proposed to guide Ca^2+^ from the membrane surface through hydrophilic clefts towards the central binding pocket [30]. It is tempting to speculate that this path, which is represented by a row of coordination sites (carbonyls) within the transmembrane protein complex, extends and connects to the surrounding annular cation–PL matrix, which supplies Ca^2+^ to the transporter via 2D exchange diffusion. We hypothesize that the unique architecture of NJs, along with the organization of membrane proteins within a specialized lipid environment, allow for exceptionally efficient and specific transfer of Ca^2+^ between sources and target sites. This proposed concept of NJ Ca^2+^ transfer combines 3D random walk of Ca^2+^ within the junctional nano-space with a process of 2D interfacial surface diffusion that feeds Ca^2+^ into the acceptor and guidance machinery of target molecules.

Hence, we propose a novel signaling concept, for which individual mechanistic steps appear plausible in view of current knowledge on the interaction of cations with biological membranes. Key steps include (i) the accumulation of Ca^2+^ ions at the surface of cell membranes by bridging the head groups of negatively charged lipids and (ii) the displacement of monovalent cations and Mg^2+^ at lipid headgroups by Ca^2+^. These molecular principles have been demonstrated by experiments in mammalian cells as well as by simulation [16,17,31]. A third step (iii), lateral flow of ions along a lipid–water interface and transport of Ca^2+^ by exchange diffusion within a phospholipid matrix, has been demonstrated in cell-free experimental systems [32]. We expect that the model proposed here can trigger research activities to test the 2D Ca^2+^ exchange diffusion hypothesis by computational as well as experimental approaches and, thereby, promote a conceptional advance towards understanding NJ Ca^2+^ signaling.

### 4.2. Implications for Human Physiopathology

This Special Issue on “Calcium Signaling in Human Health and Disease” presents burgeoning research in channelopathies and aberrant ionic signaling caused by genetic mutations, chronic disease and aging. Since the proposed process of Ca^2+^ exchange diffusion on PL membranes containing functional Ca^2+^ receptors is expected to enhance the speed and efficacy of Ca^2+^ signaling, it holds the promise of improved therapy. The specific example described in this communication focuses on refilling the ER/SR during rapid oscillatory Ca^2+^ release stimulating vasoconstriction. This mechanism has also been demonstrated in human blood vessels [33] and declined with aging in mice in parallel with a loss of NJs [34].

Dysregulation of ER Ca^2+^ homeostasis resulting in a drop of luminal Ca^2+^ is a hallmark of ER stress in chronic disease. Upregulation of rNCX to promote ER refilling, by a mechanism similar to that described in Figure 2, has recently been shown to protect primary neurons against ER stress and death in a mouse model of Alzheimer’s disease [35]. It is expected that future research of this nature on animal models, combined with physical-chemical experiments and stochastic simulations will lead to a better understanding of the mechanism whereby the components of the Ca^2+^ signaling units, or nano-junctions, interact in health and disease. Ultimately, such research can be expected to result in improvements in the management and therapy of both physical and mental diseases.

## 5. Concluding Remarks

We propose that 2D Ca^2+^ exchange diffusion on negatively charged surfaces of biological PLs, as documented earlier in an inanimate model membrane [10], functions in guiding Ca^2+^ to its receptor sites embedded on biological membranes. This constitutes a new basic component of cellular Ca^2+^ signaling; however, direct experimental or computational evidence for an involvement of this process of NJ Ca^2+^ transfer is still awaited. Nonetheless, by combining evidence from stochastic modeling and various computational as well as experimental studies with membrane models, it appears that 2D surface diffusion of Ca^2+^ at the interfaces of PL membranes and narrow aqueous phases of NJs is both feasible and essential for specific targeting of calcium-sensitive functional proteins embedded in PL bilayers. Stochastic modeling of the targeting of Ca^2+^ receptors in NJs predicts that Ca^2+^ transfer from source to target cannot be accomplished on the sole basis of 3D random walk, but becomes plausible when involving 2D exchange diffusion at the membrane’s surface, which accommodates the target. In addition, 2D interfacial Ca^2+^ diffusion may play an important role in signaling functions of other membrane systems characterized by high PL surface to aqueous volume ratios, such as diads and triads in cardiac and skeletal muscle [36]; the narrow tubular segments of the ER involved in Ca^2+^ tunnelling [37]; the narrow clefts in the nuclear envelope [38]; the luminal surfaces of the nuclear envelope itself; and the cytoplasmic surfaces of the ER containing RyRs and IP3Rs generating cellular Ca^2+^ waves [39].

## Figures and Tables

**Figure 1 ijms-23-00850-f001:**
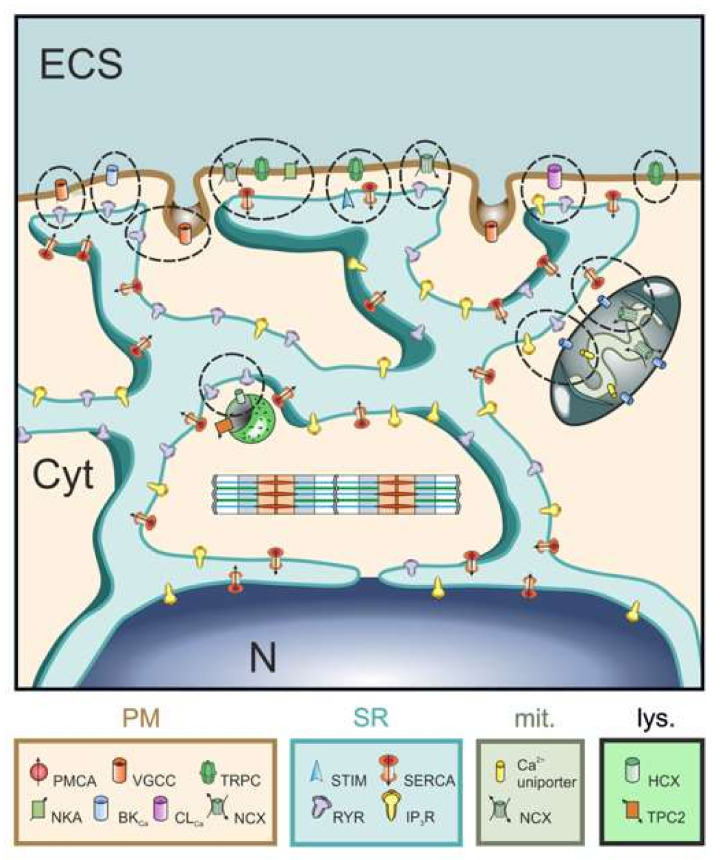
Hypothetical rendition of how the ER coordinates a multitude of different Ca^2+^ signals by involving multiple NJs between the ER membrane, on the one hand, and the PM, mitochondria and lysosomes, on the other. Each possible signaling process within NJs is indicated by dashed ellipses (based on Figure 1 in [3]). The ion transporter content of each junction (codes shown below the picture) is based on experimental evidence in the literature, which varies from solid to hypothetical. The descriptive boxes below the drawing are color-coded to their respective organelles/NJs in the drawing. ECS: extra-cellular space; cyt: cytoplasm; lys.: lysosome; mit.: mitochondria.

**Figure 2 ijms-23-00850-f002:**
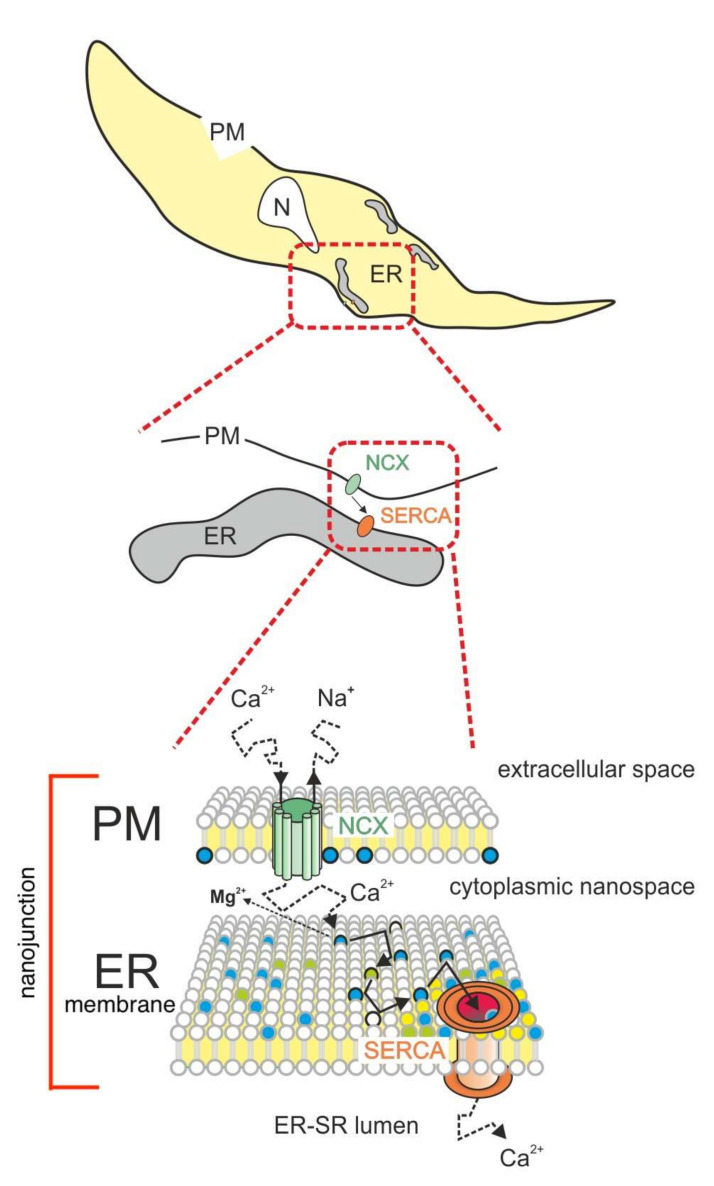
This cartoon illustrates a simplified hypothetical mechanism for refilling the ER at a NJ between the PM and ER. The proposed process of Ca^2+^ transfer between extracellular space and ER lumen is depicted in a cutout representing a distinct PM-membrane and ER-membrane-delimited nano-space (lower panel) of a cell (upper panel). Ca^2+^ enters the NJ via rNCX, where it performs a 3D random walk, and when it hits the ER surface, it proceeds by 2D exchange diffusion on the negatively charged heterogeneous PL surface of the peripheral ER. During a Ca^2+^ pulse, most of the Mg^2+^ associated with fixed coordination sites at the NJ membrane surface will be displaced into the aqueous phase. Both annular (yellow circles) and non-annular (non-yellow circles) lipids of the Ca^2+^ target protein may further facilitate guidance of Ca^2+^ into the transmembrane transport sites of SERCA.

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
