# Peer review of "Two-Dimensional Interfacial Exchange Diffusion Has the Potential to Augment Spatiotemporal Precision of Ca2+ Signaling"

_ijms, 2022, doi:10.3390/ijms23020850_

Round 1

Reviewer 1 Report

Calcium signalling at membrane contact sites (MCS) or nano-junctions (NJ) are one major focus of inter-organellar signalling field.  In the manuscript entitled “2D interfacial exchange-diffusion has the potential to augment 2 spatiotemporal precision of Ca2+ signalling”, Breemen at al proposed an intriguing hypothesis, for this unknown nature of movements of calcium ions in this highly confined space. This hypothesized “Continuous Time Random Walk” (CTRW),  or the “3D diffusion in NJ + 2D surface diffusion of calcium ions on ER membrane”,  sounds reasonable. Yet it still awaits supports from simulations.

Author Response

Reply to reviewers comments:

Reviewer 1:

Calcium signalling at membrane contact sites (MCS) or nano-junctions (NJ) are one major focus of inter-organellar signalling field.  In the manuscript entitled “2D interfacial exchange-diffusion has the potential to augment 2 spatiotemporal precision of Ca2+ signalling”, Breemen at al proposed an intriguing hypothesis, for this unknown nature of movements of calcium ions in this highly confined space. This hypothesized “Continuous Time Random Walk” (CTRW),  or the “3D diffusion in NJ + 2D surface diffusion of calcium ions on ER membrane”,  sounds reasonable. Yet it still awaits supports from simulations.

Reply: The authors thank the reviewer for her/his positive comments and appreciation of our hypothesis as both intriguing and reasonable. We agree with the reviewer that our proposal is expected to stimulate in particular computational attempts as a first test of the proposed concept.

Reviewer 2 Report

van Breemen and colleagues present, in the current manuscript, a neat hypothesis of how intracellular diffusion of Ca2+ at membrane interfaces takes place. The authors propose that Ca2+ interacts with phospholipids lining the nanojunctions between an intracellular Ca2+sink and a plasma membrane resident source to facilitate two dimensional diffusion of Ca2+ to the destined storage organelle. The authors provide elegant evidence about the mechanisms of how the charged cation interacts with the negatively charged phospholipids and support this hypothesis with existing biological examples.

This hypothesis fits well into the planed special issue and provides inspiration for future computational and mathematical models but also experimental evidence to assist understanding how diffusion of Ca2+ along phospholipid membranes improves accuracy of cellular signaling by providing another level of regulatory mechanisms.

Author Response

Reply to reviewers comments:

Reviewer 2:

van Breemen and colleagues present, in the current manuscript, a neat hypothesis of how intracellular diffusion of Ca2+ at membrane interfaces takes place. The authors propose that Ca2+interacts with phospholipids lining the nanojunctions between an intracellular Ca2+sink and a plasma membrane resident source to facilitate two-dimensional diffusion of Ca2+ to the destined storage organelle. The authors provide elegant evidence about the mechanisms of how the charged cation interacts with the negatively charged phospholipids and support this hypothesis with existing biological examples.

This hypothesis fits well into the planed special issue and provides inspiration for future computational and mathematical models but also experimental evidence to assist understanding how diffusion of Ca2+ along phospholipid membranes improves accuracy of cellular signaling by providing another level of regulatory mechanisms.

Reply: We highly appreciate the reviewer´s positive comments. Our proposal is indeed intended as well as expected to motivate computational attempts, which test this new signaling concept.

Reviewer 3 Report

This work described the 2D Ca2+ exchange-diffusion on the negatively charged phospholipid surface lining the nano-junctions participates in guiding Ca2+ from its source (channel or carrier) to its target (transport protein or enzyme). This manuscript can be considered after resolving the following points

  1. Manuscript contains several typographical and incomplete sentences, please rectified those errors??
  2. Provide more figures for better understanding the theme of the work??
  3. Discuss the mechanism?
  4. Discuss the novelty of the work??

Author Response

Reply to reviewers comments:

Reviewer 3:

This work described the 2D Ca2+ exchange-diffusion on the negatively charged phospholipid surface lining the nano-junctions participates in guiding Ca2+ from its source (channel or carrier) to its target (transport protein or enzyme). This manuscript can be considered after resolving the following points

  1. Manuscript contains several typographical and incomplete sentences, please rectified those errors

Reply: As suggested by the reviewer, we have carefully checked the manuscript and eliminated typos and incomplete sentences.

  1. Provide more figures for better understanding the theme of the work

Reply: According to the reviewer´s suggestion, we have extended the general introduction of the theme of this work (first paragraph of introduction) and include one additional illustration (new Figure 1) for better understanding of the background and scope.

  1. Discuss the mechanism
  2. Discuss the novelty of the work

Reply: As suggested, we included further discussion of individual mechanistic aspects of our hypothesis (last paragraph of discussion) and explicitly address now novelty of our concept in the discussion as well as conclusion.

The authors wish to thank the reviewer for her/his constructive comments and valuable suggestions